# Reduction of Radar Cross Section by Adopting Symmetrical Coding Metamaterial Design for Terahertz Frequency Applications

**DOI:** 10.3390/ma16031030

**Published:** 2023-01-23

**Authors:** Tayaallen Ramachandran, Mohammad Rashed Iqbal Faruque, Mandeep Singh Jit Singh, Mayeen Uddin Khandaker, Mohammad Salman, Ahmed A. F. Youssef

**Affiliations:** 1Space Science Centre (ANGKASA), Institute of Climate Change (IPI), Universiti Kebangsaan Malaysia, Bangi 43600, Malaysia; 2Centre for Applied Physics and Radiation Technologies, School of Engineering and Technology, Sunway University, Bandar Sunway, Petaling Jaya 47500, Malaysia; 3Department of General Educational Development, Faculty of Science and Information Technology, Daffodil International University, DIU Rd., Dhaka 1341, Bangladesh; 4College of Engineering and Technology, American University of the Middle East, Egaila 54200, Kuwait

**Keywords:** coding metamaterial, coding particles, lattice, radar cross section

## Abstract

This work focused on the novel and compact 1-bit symmetrical coding-based metamaterial for radar cross section reduction in terahertz frequencies. A couple of coding particles were constructed to impersonate the elements ‘0′ and ‘1′, which have phase differences of 180°. All the analytical simulations were performed by adopting Computer Simulation Technology Microwave Studio 2019 software. Moreover, the transmission coefficient of the element ‘1′ was examined as well by adopting similar software and validated by a high-frequency structure simulator. Meanwhile, the frequency range from 0 to 3 THz was set in this work. The phase response properties of each element were examined before constructing various coding metamaterial designs in smaller and bigger lattices. The proposed unit cells exhibit phase responses at 0.84 THz and 1.54 THz, respectively. Meanwhile, the analysis of various coding sequences was carried out and they manifest interesting monostatic and bistatic radar cross section (RCS) reduction performances. The Coding Sequence 2 manifests the best bistatic RCS reduction values in smaller lattices, which reduced from −69.8 dBm^2^ to −65.5 dBm^2^ at 1.54 THz. On the other hand, the monostatic RCS values for all lattices have an inclined line until they reach a frequency of 1.0 THz from more than −60 dBm^2^. However, from the 1.0 THz to 3.0 THz frequency range the RCS values have moderate discrepancies among the horizontal line for each lattice. Furthermore, two parametric studies were performed to examine the RCS reduction behaviour, for instance, multi-layer structures and as well tilt positioning of the proposed coding metamaterial. Overall it indicates that the integration of coding-based metamaterial successfully reduced the RCS values.

## 1. Introduction

Metamaterial research works have become one of the fastest-growing fields in the scientific community because they possess unique electromagnetic (EM) properties. One of the well-known applications of metamaterial is radar cross section reduction (RCS). This field is commonly adopted in military technology to produce advanced technology that is very helpful in defending a nation, such as penetrating an enemy area without being detected. The aircraft or drone with the lowest RCS values will successfully enter the foreign land, or it will be destroyed by an anti-aircraft missile if it is detected by the radars. In recent years, many research works have investigated reducing RCS values by adopting conventional metamaterial [1,2,3]. Besides that, the conventional metamaterial design is well utilized in many application fields such as specific absorption rate reduction, microwave applications, antennas, and metamaterial absorbers [4,5,6,7,8,9,10,11]. On the other hand, advanced technology is required in the current world to boost performance, thus many discoveries were introduced recently.

Therefore, research fields of coding-based metamaterial design have been commonly adopted in the past few decades as they can manipulate electromagnetic waves by arranging the unit cells’ design with distinct phase response properties. For example, some researchers proposed and investigated the coding-based metamaterial for 1-bit, 2-bit, and even 3-bits [12,13,14]. Zhao et al. created a dynamic time-domain digital-coding metasurface that makes it possible to effectively manipulate the spectral harmonic distribution [12]. Meanwhile, a review paper was produced by Zhang et al. [13] that covers the latest developments in coding metamaterial, digital metamaterial, and programmable metamaterial, along with how they might be used to build multifunctional devices and manipulate electromagnetic (EM) waves in real time. Meanwhile, Cui et al. [14] introduced digital metamaterials via two methods; for instance, proposing coding metamaterials with 0 and π phase responses with just two sorts of unit cells. Secondly, a biased diode is used to control a sole metamaterial particle with a binary response of either ‘0’ or ‘1’. In 2017, Wu et al. [15] proposed the idea of frequency coding metamaterial, which controls EM energy radiations differently when the frequency varies while maintaining a constant spatial coding pattern. Besides different phase responses, the different phase sensitivities of the unit cells are also considered in this work. Furthermore, Tran et al. [16] proposed, numerically simulated, and validated by measurement a broadband metamaterial microwave absorber. In this work, the author adopted full-wave finite integration simulation using a full-sized configuration rather than the traditional method, which is only based on unit cell boundary conditions. Initially, the coding metamaterials were started with smaller blocks of four kinds, such as 2 × 2, 3 × 3, 4 × 4, and 6 × 6, which were optimised and then used as building blocks (meta-block) for the construction of numerous 12 × 12 topologies with a realistic size scale.

Moreover, the conventional metamaterial design was also applied for terahertz frequency applications and became famous among the scientific community. In 2021, Liu et al. [17] adopted the Runge–Kutta exponential time differencing-finite difference time domain method, used to compute and simulate the RCS of the three-dimensional conductive model at terahertz frequency. Furthermore, this paper analyses the interaction between a magnetic plasma sheath and a terahertz wave. The idea of anisotropic coding metamaterial at terahertz frequencies, where the polarisation status of terahertz waves affects the coding behaviours in various directions, was proposed by Lui et al. [18]. Meanwhile, this work also experimentally exhibits an anisotropic coding metasurface that is ultrathin, flexible, and controllable by polarisation while operating in the terahertz range by adopting unique coding elements. Moreover, Iwaszczuk et al. [19] performed RCS measurements on scale models of aircraft at terahertz frequencies. More specifically, the aircraft model with a dimension of 5–10 cm was adopted in polar and azimuthal configurations in order to measure the realistic RCS value. Lui et al. [20] investigated RCS measurement through laser feedback interferometry with a terahertz quantum cascade laser in 2015.

Furthermore, conventional metamaterial structures were adopted in many research investigations to reduce RCS values. Kong et al. [21] investigated a metamaterial absorber for solar arrays with simultaneous high optical transparency and broadband microwave absorption. Analysis of light transmittance by adopting transparent substrates such as indium tin oxide film and anti-reflection glass was performed in this work. Meanwhile, Zhang et al. [22] proposed a patch antenna by loading a metamaterial absorber to reduce RCS values. The patch antenna patterns and reflection coefficient for loading and unloading the metamaterial absorber are also examined in this work. In 2014, Wen et al. [23] introduced a metamaterial absorber structure to reduce RCS values in the microwave frequency band. Besides that, the author also examined the conducting plane and dihedral angle reflectors coated with a metamaterial layer and FR-4 dielectric layer through measurement validation. On the other hand, recent developments in metamaterials and metasurfaces for RCS reduction have been reviewed; for example, basic theory, working principle, design formula, and experimental verification by Fan et al. [24].

Overall, this literature review explicitly indicates the concept of metamaterial applications in various research areas. Although many works regarding coding metamaterial were carried out in recent years, only limited studies were performed for terahertz frequency applications. Generally, extensive research works based on the coding metamaterial analysis are required to gain optimal performances in desired application fields. Thus, a few 1-bit coding sequence patterns were examined in this work by proposing compact symmetrical unit cell designs. These two types of unit cells are also known as coding particles, which act to mimic the ‘0′ and ‘1′ elements. Consequently, the element “0” coding particle was designed with only the substrate material, while element ‘1′ has a metamaterial design on the same type of substrate material. The square and symmetrical shaped metamaterial design was adopted, which was selected through the trial and error method. Additionally, two separate sets of lattices were selected for this analysis, that is, 4, 6, and 8 and 16, 20, and 24, respectively. For the smaller lattices, three different simple coding sequences were introduced and the best pattern was selected to analyse the performance in the bigger lattices. The RCS reduction values of the proposed designs were numerically calculated and the characteristics of the proposed unit cells were also analysed in this work. Meanwhile, parametric studies such as multi-layered structures and modified proposed coding metamaterial in tilt positions were performed in this work.

## 2. Material Design and Simulation

All the simulation analyses in this work were performed by adopting well-known computer software referred to as Computer Simulation Technology (CST) Microwave Studio 2019. Moreover, the simulation was run by utilising a CPU with Intel (R) Core (TM) i7-10700 @ 2.90 GHz and 16 GB RAM. The simulation takes a distinctive period to complete, which depends on the size of the structure, where smaller structures take less time when compared with the larger sized coding metamaterial. Overall, all of the simulations take approximately between the range from 21 s to 41 min. Furthermore, these simulation analyses are separated into two portions, namely properties of proposed elements (phase response and transmission coefficient) and RCS (monostatic and bistatic) simulations. These analyses adopted a time-domain solver and hexahedral mesh. The unit cell design was installed between two waveguide ports to evaluate the transmission coefficient (S21). These ports’ adopted position along the *z*-axis denotes a transverse electromagnetic mode. A perfect electric conductor was set for the *x*-axis and a perfect magnetic conductor for the *y*-axis. Moreover, the RCS reduction was successfully gained by adopting a 1-bit coding metamaterial in this work. Therefore, at the initial stage, two types of unit cells were constructed to mimic both elements in 1-bit coding metamaterial. The construction of coding particles such as ‘0′ and ‘1′ elements in coding metamaterials can manipulate the EM waves to obtain different functionalities. Then, the adopted elements are analyzed based on their phase response properties, which need to have 0° and 180° responses. Once the unit cell designs were finalized, various coding metamaterial designs with different lattices (N) such as 4, 6, 8, 16, 20, and 24 were constructed and the monostatic and bistatic RCS values were examined. A frequency range from 0 to 3 THz was used to examine the performance changes when integrating the coding-based metamaterial.

### Unit Cell Design

This research study focused on a compact and symmetrical metamaterial design by adopting a coding-based concept. The most important matter in coding metamaterial is the proposed coding elements. The selected 1-bit coding metamaterial possesses two types of unit cells such as coding elements ‘0′ and ‘1′. In this work, the coding elements were selected by adopting a trial and error method. The macroscopic medium parameters do not have to be used to characterise these ‘0’ and ‘1’ elements. It can control the behaviour of EM waves by creating coding sequences of ‘0’ and ‘1’ elements in coding metamaterials. Both unit cells in this work adopted a similar type and dimension of substrate material known as silicon. Meanwhile, element 0 only has substrate material, while element 1 is designed with a metamaterial structure by adopting copper material, as illustrated in Figure 1a,b. The metamaterial design consists of one square patch with a length and width (s) of 23 µm on the silicon material with a thickness of 0.2 µm. Furthermore, three rectangular bars with length (r1) and width (r2) of 20 µm and 1.5 µm were placed on the substrate material and subtracted from the square patch, as demonstrated in Figure 1b. The antenna and optical organizations have made extensive use of these opposite-phase techniques. The reflections of any ordinarily incident EM plane waves will cancel out if a thin, artificial perfect magnetic conductor (with 0 phase) is created and combined with perfect electric conductor (with π phase) cells in a chessboard-like layout. This will typically reduce the radar cross section values. Initially, the proposed unit cells were examined by their phase response properties, as illustrated in Figure 1c. The figure revealed that a phase difference of more than 180° was gained at two resonance frequencies, such as 0.84 THz and 1.54 THz, with a maximum response of 240° and 233°, respectively. In this case, the physical realization of coding elements is not the one that has unique properties, but it needs to possess specific responses to gain substantial phase changes to have significant freedom to control EM waves. In the binary case, the maximum phase difference is 180°. Thus, the ‘0′ element is designed as a coding element with a 0 phase response and the ‘1′ element as a 180° phase response in this work. The fact that the element ‘0′ may not have an absolute phase response of 0 at the desired resonance frequency is highlighted. This is because it can be normalized to 0 and because this case does not affect any physics. On the other hand, two types of substrate material were adopted in this analysis besides silicon material, that is, polyimide and Arlon AD410, respectively. The phase response comparison plot of these substrate materials is demonstrated in Figure 1d. It indicates that the silicon substrate material exhibits the lowest first resonance frequency among them. However, these two substrate materials have similar phase differences at distinct frequencies. Meanwhile, Figure 1e illustrates the transmission coefficient results of element ‘1′, validated by the High-Frequency Structure Simulator (HFSS) 15.0 software. CST 2019 and HFSS 15.0 simulation softwares exhibit magnitude values of −30.82 dB and −30.29 dB, respectively, at 1.8 THz. On the other hand, the full dimension details of the proposed elements are tabulated in Table 1.

## 3. Results and Discussion

The integration of coding metamaterial can manipulate EM waves and exhibit different functionalities. Coding metamaterials are directly described by quantized reflection or refraction phases, in contrast to ordinary metamaterials, which are described by continuous values of their constituent parameters. Only the phase of reflection (or refraction) is necessary in the unit cell design and system design, which greatly reduces the design process and complexity. The far-field scattering and near-field distribution are uniquely determined by the coding sequences, or coding patterns, which describe how the coding particles having different digital states are arranged in the 2D array. As a result, the researchers no longer worry about the specific structures of coding particles that make up the final coding metamaterials. The coding metamaterial analysis in this work is divided into several sections such as smaller and bigger lattices, multi-layered structure, and tilt position analysis. For the smaller size coding metamaterial analysis, three different lattices such as 4, 6, and 8 were adopted. Meanwhile, this research work also analysed slightly bigger lattices such as 16, 20, and 24, respectively, to examine the difference in RCS values. Three types of coding sequences, as tabulated in Table 2, were adopted to analyze the performance changes in monostatic RCS and bistatic RCS scattering patterns at 0.84 and 1.54 THz. These three types of coding sequences were selected through trial and methods by constructing random sequences. Moreover, the coding metamaterial structures with smaller lattices adopted the same sequence patterns as demonstrated in Table 2, while the bigger lattices only analysed the best sequence from these three. The results of the smaller lattices are demonstrated as shown in Figure 2 and Table 3 and Table 4. Figure 2a–c also demonstrate the monostatic RCS values and the results revealed that the increase in the number of lattices causes the RCS values to decrease besides the reduction behaviour by adopting various coding sequences. On the other hand, coding sequence 2 has the lowest bistatic RCS values for all types of sequences in 4, 6, and 8 lattices, as illustrated in Table 3 and Table 4. For instance, the coding metamaterial design with 8 lattices exhibits the lowest bistatic RCS value of −65.5 dBm2 at 1.54 THz. Furthermore, the increasing number of lattices exhibit distinct bistatic RCS reduction differences from 4 to 8 lattices; for example, 8.2, 9.8, and 12.1 at 0.84 THz. Meanwhile, the proposed coding sequences exhibit differences of 5.3, 4.3, and 3.9 at 1.54 THz. The analysis of three coding sequences indicated that the manipulation of EM waves is possible by simply arranging the proposed unit cells with 0° and 180° phase responses in the desired lattice form.

### 3.1. Larger Lattices

On the other hand, bigger lattices were analysed by adopting the coding sequence 2 pattern in this parametric study. It is clearly shown in Figure 3 that the RCS values simultaneously reduced when the number of lattices was increased. For example, Table 5 demonstrates bistatic RCS scattering patterns that reduce from −57 dBm^2^ to −50.2 dBm^2^ and −49.3 dBm^2^ to −42.3 dBm^2^ at 0.84 THz and 1.54 THz, respectively. Thus, this indicates that the RCS values can be minimised by adopting bigger lattices and, for miniaturisation constraints, anyone can modify the coding sequences to gain distinct performances.

### 3.2. Multi-Layer Coding Metamaterial

In this parametric study, a similar proposed coding sequence 2 was adopted to analyse the effect of multi-layered properties on the RCS reduction applications. The multi-layered metamaterial design structure is applied in many application fields such as absorption reduction, filter, wireless communications, and so on. This unique property has great potential in exhibiting not only single charcacteristics, but plenty of characteristics. Therefore, the 4 and 16 lattices with coding sequence 2 designs were selected for this analysis. Table 6 illustrates the bistatic scattering patterns of the proposed coding metamaterial with varied layers such as 1, 2, 3, and 4, respectively. It is clearly revealed that the increasing number of layers, up until 3, exhibits an appropriate reduction in RCS values, for instance, successfully decreasing from *−*78.6 dBm^2^ to *−*69.7 dBm^2^ and *−*57 dBm^2^ to *−*50.9 dBm^2^ at 0.84 THz for 4 and 16 lattices, respectively. Meanwhile, the 4 layers coding metamaterial design manifests slightly increasing RCS values instead of reducing in both lattices. In a nutshell, the reduction of RCS values can be gained by adding multiple layers based on the performance and desired applications.

### 3.3. Tilt Position

For the final parametric study, the proposed coding sequence 2 design with 8 lattices was adopted to analyse the changes in performances by tilting it at various angles such as 5°, 10°, 25°, and 50°, as illustrated in Figure 4. Meanwhile, the coding metamaterial in tilt position does not exhibit the desired reduction behaviour and only increases when the angle varied along the *x*-axis. However, the changes in the angle of the tilt position provide freedom to change the scattering patterns, as demonstrated in Table 7. The scattering patterns for the coding metamaterial at 5° illustrate distinctive behaviour, while it changes the direction when the angle of tilt position increases. This will be beneficial for those looking to gaining controllable scattering patterns in the distinct frequency range by only manipulating the angle of tilt position. The bistatic scattering patterns of this analysis are tabulated in Table 7.

Table 8 illustrates the comparison of a few research works in terahertz frequency regimes that adopt various types of metamaterial or metasurface for example [25,26,27,28]. All the research works utilised a frequency range below 5 THz and have distinct sizes. In the double layer structure category, the proposed coding metamaterial has the smallest design, while the work of [26] constructed it with a slightly bigger design, likely 200 µm × 100 µm reconfigurable metasurface. Meanwhile, the monolayer designs such as those in [27,28] possess a compact structure compared with the rest of the designs. In each research work, the authors successfully investigated by adopting a specific metamaterial type for desired applications. Overall, constructing a compact coding metamaterial justifies the unique behaviours in this work.

## 4. Conclusions

This work provided a novel compact symmetrical coding metamaterial design and effectively accomplished the analysis of their behaviours on the RCS reduction values. Two different types of unit cells with comparable size and substrate materials were proposed because the study employed a 1-bit coding metamaterial. In contrast, the analytical simulation showed that, when the changes in coding sequences occur and the number of lattices rises, the RCS values can be decreased by adopting a coding-based metamaterial design. This proves that the coding metamaterials simply control EM waves through various coding sequences of ‘0’ and ‘1’ components, as opposed to the existing analogue metamaterials, which employ effective medium parameters or unique dispersion relations to control EM fields. For example, the smaller coding metamaterial such as 4, 6, and 8 lattices, which have sequence patterns of 1010,…/101010,…/10101010,…, exhibit changes in monostatic and bistatic RCS values at 1.54 THz. Meanwhile, the larger coding sequences such as 16, 20, and 24 lattices, which adopted the coding sequence 2 pattern, exhibit excellent scattering patterns. Finally, the multi-layered coding metamaterial also exhibits unique changes in bistatic scattering patterns and reduces monostatic RCS from −57 dBm^2^ to −53 dBm^2^ when increasing the layers at 0.84 THz. In contrast, each lattice allows for the simple manipulation of EM waves by arranging the unit cells in a particular order. In summary, the coding metamaterial has the potential to be used in a wide range of applications such as metamaterial absorbers, lenses, cloaking structures, and so on. This is because it allows the manipulation of EM waves through the use of both 1-bit coding elements in a straightforward sequence arrangement.

## Figures and Tables

**Figure 1 materials-16-01030-f001:**
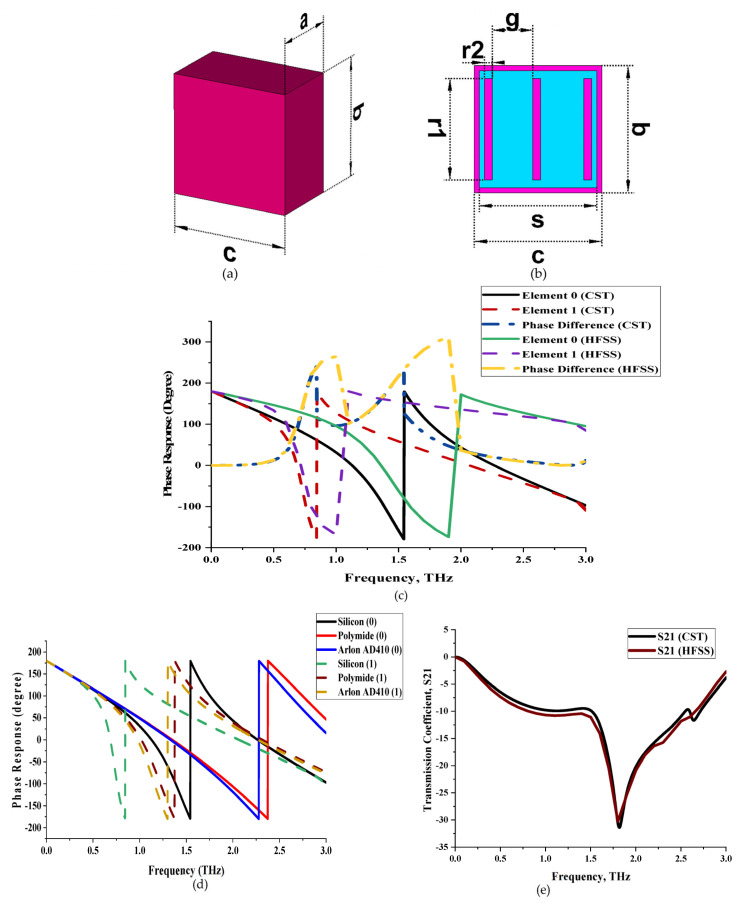
Unit cells of coding metamaterial: (**a**) element 0; (**b**) element 1; (**c**) phase responses from CST and HFSS; (**d**) phase response in various substrate material; (**e**) transmission coefficient.

**Figure 2 materials-16-01030-f002:**
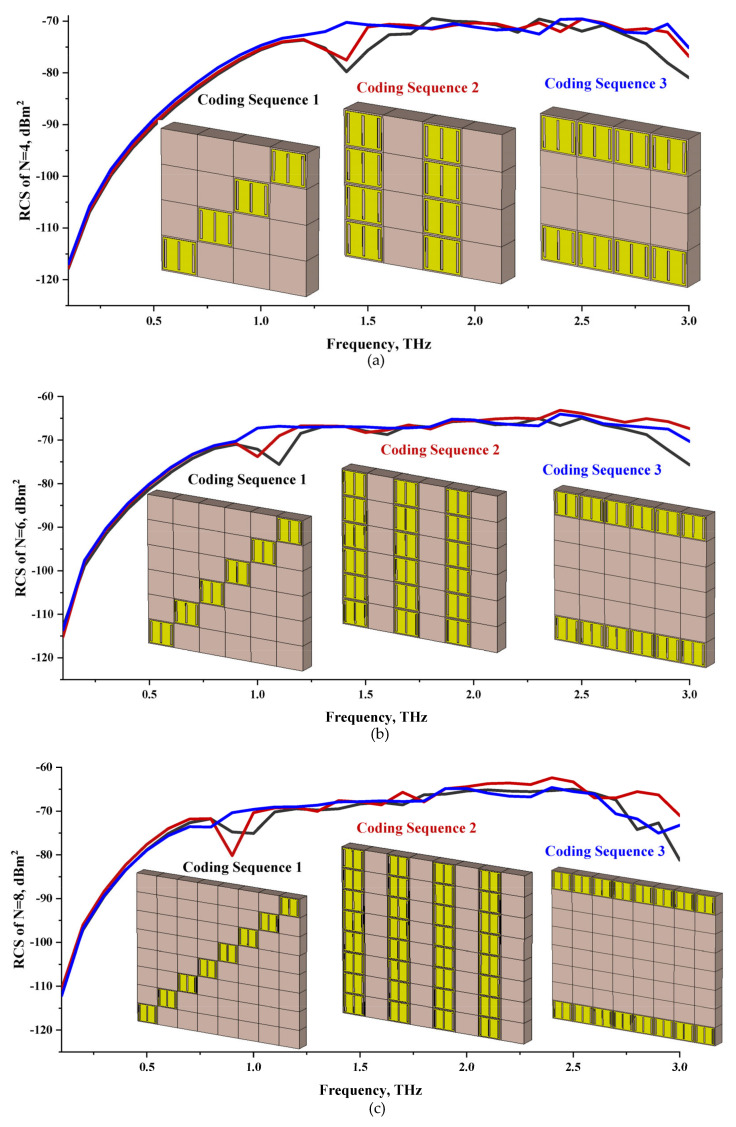
Monostatic RCS reduction of three different coding sequences in (**a**) N = 4; (**b**) N = 6; (**c**) N = 8.

**Figure 3 materials-16-01030-f003:**
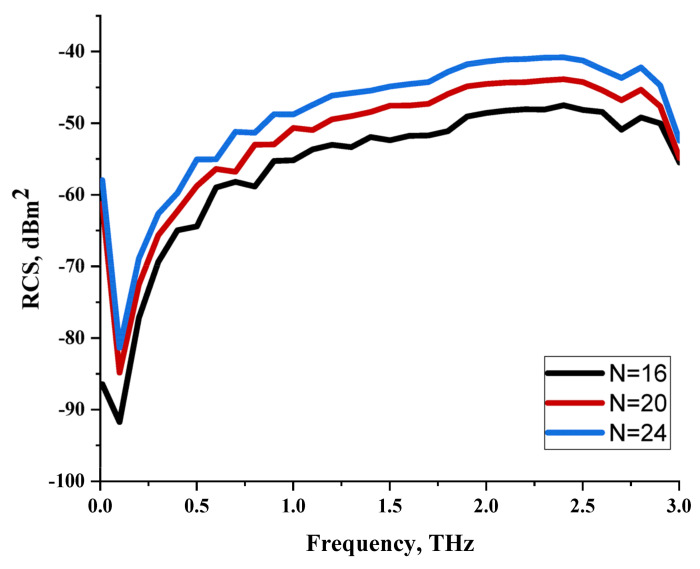
Monostatic RCS reduction of three different coding sequences in N = 16, N = 20, N = 24.

**Figure 4 materials-16-01030-f004:**
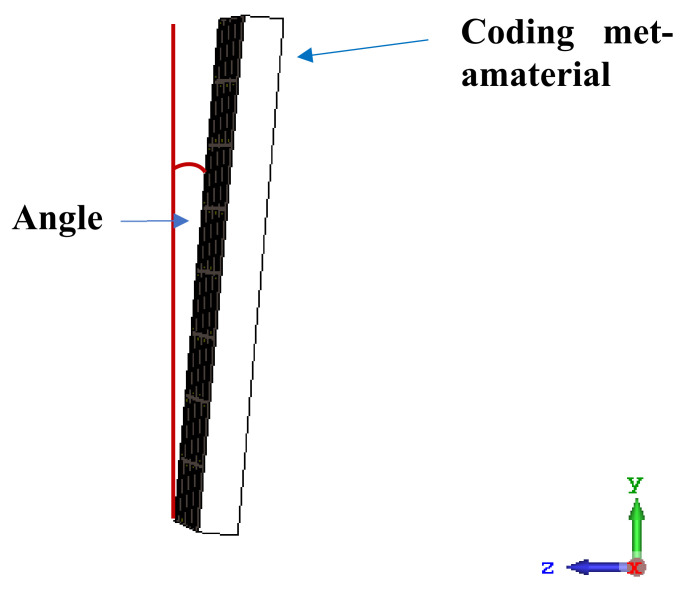
Sample of the tilted position of the proposed coding metamaterial.

**Table 1 materials-16-01030-t001:** Dimension of both elements.

Descriptions	Dimension (µm)
a	15
b	25
c	25
s	23
r1	20
r2	1.5
g	8.0

**Table 2 materials-16-01030-t002:** Coding sequences pattern for 4 lattices.

Coding Sequence	Row 1	Row 2	Row 3	Row 4
1	1000	0100	0010	0001
2	1010	1010	1010	1010
3	1111	0000	0000	1111

**Table 3 materials-16-01030-t003:** Bistatic RCS scattering patterns of three types of lattices at 0.84 THz.

**Lattices**	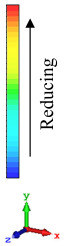	**Coding Sequence 1**	**Coding Sequence 2**	**Coding Sequence 3**
4		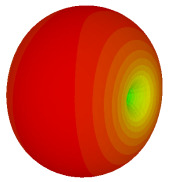	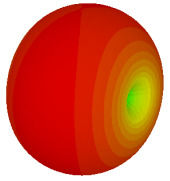	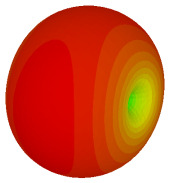
		−79 dBm^2^	−78.6 dBm^2^	−77.7 dBm^2^
6		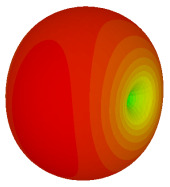	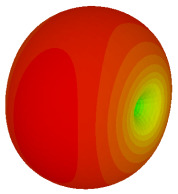	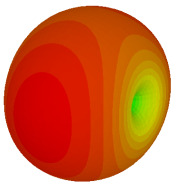
		−71.3 dBm^2^	−70.7 dBm^2^	−70.3 dBm^2^
8		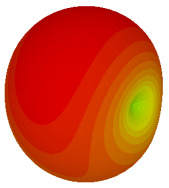	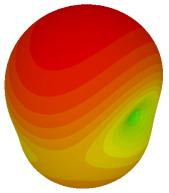	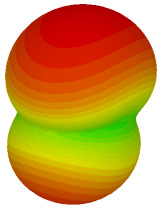
		−70.8 dBm^2^	−68.8 dBm^2^	−65.6 dBm^2^

**Table 4 materials-16-01030-t004:** Bistatic RCS scattering patterns of three types of smaller lattices at 1.54 THz.

**Lattices**	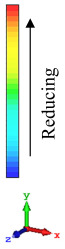	**Coding Sequence 1**	**Coding Sequence 2**	**Coding Sequence 3**
4		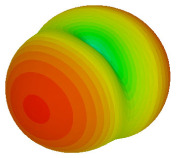	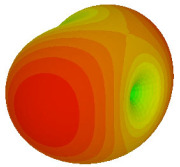	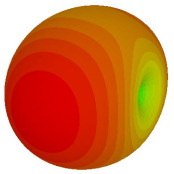
		−71.6 dBm^2^	−69.8 dBm^2^	−71 dBm^2^
6		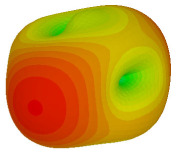	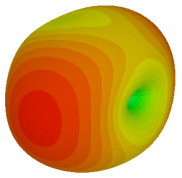	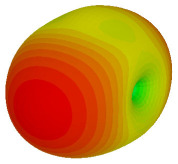
		−66.8 dBm^2^	−67.8 dBm^2^	−67.1 dBm^2^
8		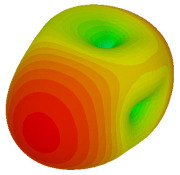	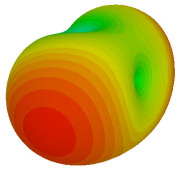	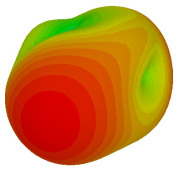
		−66.3 dBm^2^	−65.5 dBm^2^	−67.1 dBm^2^

**Table 5 materials-16-01030-t005:** Bistatic RCS scattering patterns of three types of bigger lattices at 0.84 and 1.54 THz.

**Lattices**	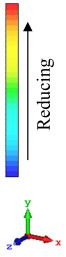	**16**	**20**	**24**
0.84		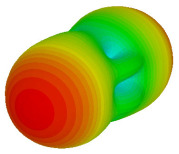	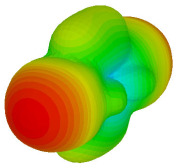	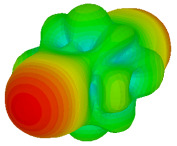
		−57 dBm^2^	−52.8 dBm^2^	−50.2 dBm^2^
1.54		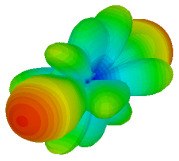	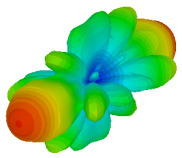	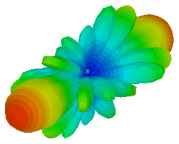
		−49.3 dBm^2^	−45.7 dBm^2^	−42.3 dBm^2^

**Table 6 materials-16-01030-t006:** Bistatic RCS scattering patterns of coding metamaterial with 4 and 16 lattices at 0.84 THz.

**Layer**	**Smaller Lattices**	**Bigger Lattices**
1	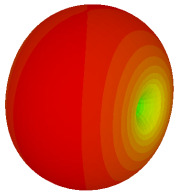	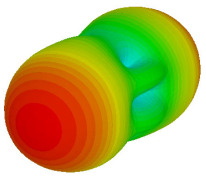
	−78.6 dBm^2^	−57 dBm^2^
2	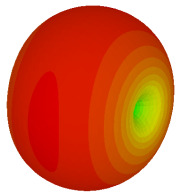	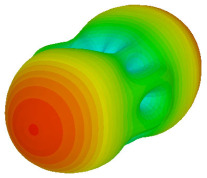
	−73.7 dBm^2^	−55 dBm^2^
3	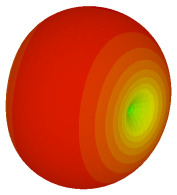	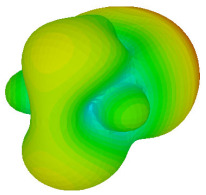
	−69.7 dBm^2^	−50.9 dBm^2^
4	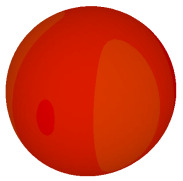	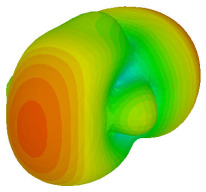
	−74.9 dBm^2^	−53 dBm^2^

**Table 7 materials-16-01030-t007:** Bistatic RCS scattering patterns of coding metamaterial with 8 lattices at 0.84 THz with various tilt positions.

Angle	Bistatic RCS
5	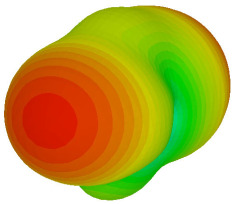
	−61.8 dBm^2^
10	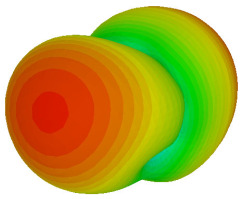
	−62.7 dBm^2^
25	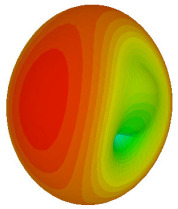
	−69.6 dBm^2^
50	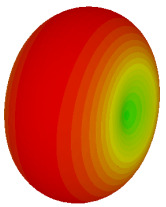
	−70.5 dBm^2^

**Table 8 materials-16-01030-t008:** Comparison of previous research works.

Reference	Frequency (THz)	Dimension (µm)	Structure	Type
[25]	0.7 to 1.3	50 × 50	Double layer	Anisotropic coding metamaterial
[26]	0.25 to 0.65	200 × 100	Double layer	Reconfigurable metasurface
[27]	3 to 4.2	12 × 12	Monolayer	Graphene metasurface
[28]	0.8 to 1.6	14 × 14	Monolayer	Graphene coding metamaterial
Proposed	0 to 3	25 × 25	Double layer	Coding metamaterial

## Data Availability

All the data are available within the manuscript.

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
