# Peer review of "Reduction of Radar Cross Section by Adopting Symmetrical Coding Metamaterial Design for Terahertz Frequency Applications"

_materials, 2023, doi:10.3390/ma16031030_

Round 1
Reviewer 1 Report
Some results are presented based on the design device. I suppose that more physical mechanism can be given to improve the scientific value for the manuscript.
Comments,
(1) In the results and discussion, more physical mechanism should be given for each phenomenon to improve the scientific nature.
(2) It is difficult to rightly judge the presented simulation results, so, can you give some comparation with other reported results (experimental results).
(3) Can you give some potential applications for the presented device.
Author Response
As attached.

Reviewer 2 Report
Comments on the manuscript
In this manuscript entitled “Reduction of radar cross section by adopting symmetrical coding metamaterial design for terahertz frequency applications,” the authors presented a systematic study on coding metasurfaces for the purpose of radar cross-section reduction. Specifically, they shrink the meta-atom size and make their devices operate in THz. Moreover, they also include finite size discussion and oblique incidence discussion on their coding metasurfaces. The manuscript, in general, is interesting and their simulation results are incremental to the field of coding metasurfaces. Besides, the manuscript is technically sound with well-supported conclusions and assertions. However, the language is a little difficult to understand with some grammar errors, which need to be improved in the revised manuscript.
Thus, this manuscript meets the scope of Materials. My comments and suggestion to the authors are listed below.
1. My major concern is related to the novelty of implementing radar cross-section reduction in the THz range using compact meta-atom. Coding metamaterials have been well explored in many aspects, which are often in the microwave regime [Ramachandran, Tayaallen, et al. "Coding Metamaterial Analysis Based on 1-Bit Conventional and Cuboid Design Structures for Microwave Applications." Materials 15.21 (2022): 7447]. But I am unable to recognize this manuscript's remarkable improvement. Thus what is the key selling point or unique advantage for the THz metasurfaces with a compact lattice spacing? The authors need to think seriously about these comments and questions, and briefly highlight the selling point in the abstract since it is closely related to the technical innovations and scientific impact of the manuscript.
2. “Besides that, the conventional metamaterial design is well utilized in many application filed such as Specific Absorption Rate reduction, microwave applications, antennas, etc. [4]-[9]” Metamaterial absorber is an important direction of metamaterials. Better to include this aspect as well. These two works may help you. [Liang, Yao, et al. "Hybrid anisotropic plasmonic metasurfaces with multiple resonances of focused light beams." Nano Letters 21.20 (2021): 8917-8923; Pati, Shyam Sundar, and Swaroop Sahoo. "Single/Dual/Triple Broadband Metasurface Based Polarisation Converter with High Angular Stability for Terahertz Applications." Micromachines 13.9 (2022): 1547].
3. In “3.2. Tilt Position”, “Meanwhile, the coding metamaterial in tilt position does not exhibit the desired reduction behaviour and only increasing when the angle varied along the z-axis.” First, grammar errors. Should be “…only increases…” Second, the bistatic RCS for various tilted angles seems low. Indeed, angular stability is an important criterion of broadband metamaterials absorbers and coding metamaterials with strong radar cross-section reduction. How did the authors interpret their claim of “However, the changes in the angle of the tilt position provide great freedom to control scattering patterns.” Some samples for this claim would help.
Author Response
As attached.

Reviewer 3 Report
The most important weakness of this work is the absence of any theoretical analysis that can support the claims of the authors. In this way the paper seems to be only an application of metamaterials to specific action.
1. The analysis of Section 2 is rather limited. What is the motive of the authors and particularly the methodology for selecting the specific material pattern? Have they compared it with other structures prior to concluding to this?
2. In the results section the results are not so informatively presented, namely the surface plots are a lot. Please try to make them more compact and then place comments that could help the readers.
3. Moreover, there are not any comparisons with other implementations or existing schemes.
4. The computational evidence should be definitely provided, i.e. CPU time and RAM for the implementations.
5. The quality of all 2-D plots should be improved.
Author Response
As attached.

Round 2
Reviewer 1 Report
After revision, the manuscript has been improved. So, It can be accepted now.